# Three-Dimensional Planning and Patient-Specific Instrumentation for the Fixation of Distal Radius Fractures

**DOI:** 10.3390/medicina58060744

**Published:** 2022-05-30

**Authors:** Tatjana Pastor, Ladislav Nagy, Philipp Fürnstahl, Simon Roner, Torsten Pastor, Andreas Schweizer

**Affiliations:** 1Department of Hand Surgery, Balgrist University Hospital, University of Zurich, 8008 Zurich, Switzerland; ladislav.nagy@balgrist.ch (L.N.); simon.roner@ksgr.ch (S.R.); torsten.pastor@luks.ch (T.P.); andreas.schweizer@balgrist.ch (A.S.); 2Department of Plastic and Hand Surgery, Inselspital University Hospital Bern, University of Bern, 3007 Bern, Switzerland; 3Computer-Assisted Research and Development Team, Balgrist University Hospital, University of Zurich, 8008 Zurich, Switzerland; philipp.fuernstahl@balgrist.ch

**Keywords:** distal radius fracture, 3D planning, patient-specific guide, preoperative planning, osteosynthesis

## Abstract

*Background and Objectives*: Three-dimensional planning and guided osteotomy utilizing patient-specific instrumentation (PSI) with the contralateral side used as a reference have been proven as effective in the treatment of malunions following complex fractures of the distal radius. However, this approach has not yet been described in relation to fracture reduction of the distal radius. The aim of this study was to assess the technical and logistical feasibility of computer-assisted surgery in a clinical setting using PSI for fracture reduction and fixation. *Materials and Methods*: Five patients with varied fracture patterns of the distal radius underwent operative treatment with using PSI. The first applied PSI guide allowed specific and accurate placement of Kirschner wires inside the multiple fragments, with subsequent concurrent reduction using a second guide. *Results*: Planning, printing of the guides, and operations were performed within 5.6 days on average (range of 1–10 days). All patients could be treated within a reasonable period of time, demonstrating good outcomes, and were able to return to work after a follow-up of three months. Mean wrist movements (°) were 58 (standard deviation (SD) 21) in flexion, 62 (SD 15) in extension, 73 (SD 4) in pronation and 74 (SD 10) in supination at a minimum follow-up of 6 months. *Conclusions*: Three-dimensional planned osteosynthesis using PSI for treatment of distal radius fractures is feasible and facilitates reduction of multiple fracture fragments. However, higher costs must be taken into consideration for this treatment.

## 1. Introduction

The primary aims in the operative treatment of displaced intra-articular distal radius fractures are anatomical reduction of the articular surface and restoration of normal alignment. Several authors have suggested that a persisting intra-articular step of more than 2 mm may lead to radiocarpal degeneration, wrist stiffness and functional deficits [1,2,3]. The wrist and forearm function as a unit; therefore, failure to sufficiently restore alignment at the distal radius can lead to radioulnar dysfunction and decreased range in pronation/supination [4,5,6]. In the setting of multiple articular fragments, displacement in multiple directions can be caused by forces resulting from the carpus and the soft tissue attachments of the capsule, carpal ligaments, and brachioradialis. The concurrent control and reduction of multiple fragments, maintenance of reduction, and subsequent fixation, while avoiding interference with implant positioning, can be surgically challenging, especially in poor bone quality or small fragments.

Although there have been important developments in the surgical technique and implant function, long-term results still demonstrate high rates of arthrosis [1,7]. The accurate restoration of the articular surface of both radiocarpal and distal radioulnar joints is paramount in prevention of post-traumatic osteoarthritis [8]. This is especially important in young patients with distal radius fractures. The use of computer-assisted orthopaedic surgery (CAOS) in trauma patients has the potential to improve both intraoperative workflow and fracture reduction quality, which may prevent postoperative arthritis and improve functional results. Studies have concluded that virtual preoperative planning and three-dimensional (3D) printing devices used for shoulder and pelvis operative treatment can improve surgeon's satisfaction, reduce operative time and radiation exposure, and even improve patient outcomes [9,10]. In the distal radius anatomical region, several studies on malunions reported accurate reduction of the intra-articular surface via corrective osteotomies guided by 3D-printed patient-specific instruments (PSIs) [11,12,13]. Utilizing this technique, it can be easier and faster to perform complex osteotomies, being difficult to set when using conventional techniques. The process utilizes computed tomography (CT) scans of the affected and contralateral sides, which are segmented and reconstructed to create 3D virtual models. The contralateral side can be mirrored and used as a normal template, the osteotomy can be planned virtually, and the 3D-printed guides can be utilized intraoperatively. The operative PSIs are designed to follow the anatomical contours of the bone, and only fit properly in an appropriate position—specific Kirschner (K-) wire paths can be designed to allow proper positioning around the metal hardware. This approach has been utilized successfully in some previous studies on the upper extremity [11,12,13]. However, no studies exist yet that assess the utility of PSI for treatment of complex distal radius fractures, where the potential for control and fixation of multiple fragments may improve the accuracy of reduction and enhance the operative workflow. In the setting of multifragmentary intra-articular fractures, surgeons often use adjuncts, such as arthroscopy and extended or additional approaches, for visualization. An intraoperative aid that accurately addresses multiple fragments may negate the need of using such additional invasive aids.

Therefore, this study sought to validate the feasibility of a process of preoperative virtual reduction involving the design and use of PSI as an intraoperative aid for treatment of complex distal radius fractures.

The indications for this technique are complex distal radius fractures that do not necessitate immediate surgical intervention. Contraindications are fractures requiring emergent treatment, such as contaminated open fractures, or those with nerve or vessel compromise. Fractures initially treated with an external fixator are ideal for this approach due to the staged conversion to internal fixation with sufficient time for preoperative planning. In the light of higher costs and time expenditure, this technique is not indicated for simple but for complex multifragmentary intra-articular and extra-articular fractures. This demonstrates the utility and flexibility of the technique for a heterogenous group of both fracture types and fragment-specific fixations.

## 2. Materials and Methods

Ethical approval for this feasibility study was granted by the institutional review board (Kantonale Ethikkommission Zürich, Switzerland, BASEC-Nr. 2019-00369 for clinical cases) and all included patients were consented for their specific fracture treatment. Five patients with distal radius fractures underwent CT imaging, preoperative planning and intraoperative use of PSI during the treatment.

### 2.1. Included Patients and Fracture Type

The ages of patients, AO/OTA fracture types and implant data are presented in Table 1.

The first case was presented at our clinic after a fall at work four days before. Patient history included a previous operatively treated intra-articular distal radius fracture two years before, with subsequent implant removal a year later. After the initial trauma, there was incongruency at the scaphoid fossa and the radial inclination was reduced versus the contralateral side. The fracture was extra-articular.

The second patient fell on an outstretched hand and was presented 10 days later to our outpatient clinic. Imaging showed a large volar Barton fragment and a depressed fragment of the central radial styloid.

The third patient arrived at our clinic after falling down the stairs. A CT scan demonstrated a coronal intra-articular split with displacement of the dorsal articular segment, dorsal comminution, and a radial styloid fragment.

The fourth patient suffered a grade 1 open distal radius fracture. Closed reduction and external fixation were performed prior to referral to our hospital, accompanied by ongoing significant soft tissue swelling. A metaphyseal wedge fragment was present with significant translation and tilt of the distal fragment.

The fifth case was from a mountain bike accident, with the patient suffering multiple concomitant hand fractures. The distal radius fracture included a displaced partial articular radial styloid fragment, also known as a Chauffer fracture.

### 2.2. Preoperative Planning

All patients received a CT scan (slice thickness 1 mm; 120 kV; Philips Brilliance 40 CT, Philips Healthcare, The Netherlands) of the injured an contralateral forearms from the elbow past the radiocarpal joint (see Table 2 for CT protocol details). The CT scans of bilateral radii were segmented using commercial software (Mimics V.20.0, Materialise, Leuven, Belgium) and reconstructed to form triangular surface models. To perform segmentation, manual thresholding and region growing were applied. The mirrored contralateral intact side served as a template. Computer planning was performed by the senior surgeon using the custom-made software application CASPA (Balgrist CARD AG, Zurich, Switzerland). The fracture fragments were aligned with the intact contralateral side using an iterative closest point (ICP) surface registration algorithm. A 3D model of the planned locking plate for operative fixation was positioned on the bone surface (Figure 1). Cylinders represented the exact position of the angular-stable locking screws in the plate model. Finally, PSIs were produced by means of an in-house 3D printer (Formiga P100, EOS GmbH, Krailling, Germany). The extreme fine-focus diameter enabled a wall thickness of 0.4 mm at a slice thickness of 100 µm and a printing speed of 24 mm/h in height. The guides were made from polyamide and sterilized with a conventional steam pressure sterilizer. Cleaning and sterilization were performed according to the cleaning and sterilization instructions of MyOsteotomy (Medacta SA, Castel San Pietro, Switzerland). Equivalency to this approved medical product was demonstrated and the necessary documentation was provided to the respective regulatory bodies. Correspondingly, a regulatory approval for use of the printer was granted for the purpose of the study. General licensing was planned as a subsequent step, with outsourcing to a MedTech company. The printing material was PA2200, a commonly used material for patient-specific instruments. It was validated as class VI (the highest class) by the United States Pharmacopeia (USP). Surgeons and engineers with expertise in 3D planning of corrective osteotomies performed the planning, with approximate time effort of 2–3 h. The duration of the printing was dependent on the number of needed parts and lasted several hours, including the cooling time for the printer. A printing time of 8–12 h was estimated for a distal radius fracture. The overall costs were approximately 1500 EUR for planning and design of the PSI, and 600 EUR for PSI production, including material costs of approximately 200 EUR.

Two types of PSI guides were needed to reduce a fracture. The first one was a fragment-specific fixation guide (Figure 1, Case 1, Image 3). The PSI was designed to closely articulate the specific articular fragments in displaced position and allow accurate K-wire placement. The shape was designed based on the anatomical cortical surface and incorporated the shape of the fracture line (Figure 2). The guide was designed based on the imaging of the fractured bone in a displaced position. Although fractures may further dislocate after initial CT, the PSI shape with its individual facets for each fragment was such that when applied to the bone with pressure, the fragments could return to their previous relative positions. The second PSI was a reduction aid (Figure 1, Case 1, Image 4) attached to the locking plate via K-wire guides in line with the locking holes of the plate. When the K-wires were accurately placed in each fragment, passing them into the reduction aid resulted in anatomical alignment of the fracture fragments. The wires could then be replaced with locking screws. After using the first guide for K-wire placement, the reduction aid could be applied with a plate (Case 1), alone (Case 2), or alternatively the plate alone could be used for positioning (Case 3). Both preoperative segmentation of the CT scans of the affected and contralateral side and preoperative planning were performed within 1–3 days after the first clinical consultation at our hospital. Urgency was dictated by planned procedure date, and no procedures were pushed back due to surgical planning.

### 2.3. Operative Management

Surgery in all patients proceeded after adequate resolution of the soft tissue swelling and was performed by senior hand surgeons (A.S. in four cases and L.N. in one case). The modified Henry approach to the distal forearm was utilized for all patients. Either 2.4 LCP Distal Radius Plate (Synthes, Zuchwil, Switzerland) or Correctus Plate (Intercus, Aarau, Switzerland) were used according to surgeon preference. Each plate was positioned with the use of the corresponding PSI and then fixed (Figure 3 and Video S1). Reduction was confirmed via intraoperative fluoroscopy (Siemens ARCADIS Varic, Siemens Medical Solutions AG, Erlangen, Germany).

Postoperatively, the range of motion was measured and the grip strength was evaluated with a dynamometer (JAMAR; Sammons Preston, Bolingbrook, IL, USA). Clinical follow-up of all patients was performed after two weeks, six weeks, three months, and six months. Two patients were inspected again after one year. Radiographic review was performed after six weeks and six months. All patients received a volar slab for two weeks which was replaced with a removable cast for further four weeks.

## 3. Results

### 3.1. Patient Characteristics

The patient group consisted of three cases with an intra-articular fracture and two cases with an extra-articular fracture, comprising three male and two female patients. The mean age was 57 (range 30–72) years, the mean duration until surgery was 16.8 (range 3–21) days after trauma and 5.6 (range 1–10) days after presentation to our clinic. One case had an extended time to surgery due to late presentation after injury abroad, with significant soft tissue swelling.

### 3.2. Preoperative Planning

The PSI production, including CT analysis and virtual planning, took approximately 8 h to complete, considering one hour effort of the treating surgeon and further commitment of an in-hospital engineer. The PSI was 3D-printed overnight in all cases, being ready by the next morning.

### 3.3. Clinical Results

At the final follow-up all patients demonstrated good functional results, with none of the patients reporting wrist pain (Table 3). All patients were able to return to work after three months (Table 2). The radiological review revealed union and maintenance of the standard alignment parameters on X-ray (radial height, inclination, and volar tilt) at 6–8 weeks. At the final follow-up none of the patients demonstrated degenerative changes, such as subchondral sclerosis, joint space narrowing or osteophyte formation. No complications were observed. Implant removal was performed in one patient due to soft tissue irritation.

## 4. Discussion

The current study demonstrated a feasible end-to-end process within the hospital facilities—from patient imaging, segmentation and preoperative planning to fixation using patient-specific intraoperative guides—being useful for a wide range of fracture types across the spectrum of the AO/OTA classification.

Studies utilizing CAOS have reported excellent results in patient outcomes and accuracy of reduction [3,11,14]. CAOS has been applied to various malunions of the upper limb in the forearm and extra-articular distal radius, as well as intra-articular malunions of the distal radius. A more accurate reduction has been reported compared to freehand techniques [2,15,16]. In the current study we introduced this technique for use in distal radius fractures and presented five varied clinical cases. Similar to the positioning for intra-articular malunion correction, the jig had an articulating face to hold each fracture fragment based on the preoperative CT. Although the fragments were more mobile and prone to displacement, the multifaceted jig, when pushed at the fracture site, allowed each fragment to fall into the planned position for K-wire fixation. This technique was applied to simple and multifragmentary metaphyseal extra-articular fractures. It also demonstrated utility for fragment-specific fixation in a partial articular styloid fracture, as well as control and fixation in complex multifragmentary intra-articular fractures including a volar Barton’s fragment. The method is thus versatile and applicable to a variety of fracture types.

Complex comminuted distal radius fractures or fractures with multiple fragments in the metaphysis are challenging to reconstruct. The fragments are displaced in different directions due to their soft tissue attachments and the original traumatic impact and orientation can be difficult when anatomic bone landmarks (e.g., volar rim of the distal radius) are displaced. There has been a trend to utilize intraoperative arthroscopic assistance in such complex intra-articular cases [17,18,19,20,21,22]. Treating such complex fractures remains challenging due to limited access to the articular surface. Some surgeons perform extended additional approaches or arthrotomy for improved visualization. However, an extensive soft tissue dissection to access the joint may increase the risk of postoperative scarring and stiffness. Even in cases with full joint visualization, the accurate reduction and concurrent fixation of multiple mobile fracture fragments remains challenging [23,24].

It is important to accurately restore the radiocarpal joint surface after trauma, as a disturbance in gliding due to postoperative gaps or steps in the joint surface leads to osteoarthritis, as demonstrated in several studies [1,25,26]. Furthermore, a fourfold increase in the joint load was reported after a joint step of 1 mm and an eightfold increase after a joint step of 2 mm in a biomechanical cadaver study [27]. This is in line with several clinical reports. A strong correlation between residual joint congruence with a step-off larger than 1.5 mm and the development of arthrosis was reported after a minimum follow-up of five years [25]. Furthermore, a postoperative step-off of more than 1 mm was demonstrated in 28% of patients treated with the conventional technique with open reduction and internal fixation [28]. These clinical and biomechanical studies underline the importance of accurate joint surface restoration. Schweizer et al. presented six patients after correction osteotomy for distal radius malunions using a similar 3D PSI operation technique. The infrastructure used (printer, software, technicians) was identical to the infrastructure of the current study. Although they did not perform on acute fractures, a joint surface congruency restoration below 1 mm was achieved [3]. However, whether those values may be achieved with the described technique of the current study needs to be further evaluated. Moreover, additional research is necessary regarding the clinical outcomes of larger patient collectives operated on with the presented technique. The use of PSI was demonstrated as useful for accurate K-wire positioning, allowing for concurrent reduction and control of all fragments. If accurate articular reduction can be performed consistently in bigger patient cohorts, it may negate the need for more invasive articular access.

This technique has several limitations. The use of a contralateral CT scan as a template exposes the patient to further radiation. An alternative in the future may be to use a statistical shape model representing a population-wide variation in anatomical shapes [25]. Furthermore, the planning, design, and production of the PSI are expensive, and the time requirement from both a surgeon and an available engineer with software access is not negligible. An ongoing critique of many examples of computer-assisted orthopaedic surgery is that the time and cost investment are currently disproportionate to the demonstrated benefits. While this may reflect the current situation, the research in image segmentation and virtual surgical planning is advancing at a rapid pace. One of the senior authors has already published methods in automated segmentation and virtual fracture reduction [26,27,28]. We believe that in the next decade the advances in image processing and 3D printing will allow for faster and cheaper application of computer-assisted surgery. Therefore, it is important to establish potential clinically beneficial methods and pipelines in the present that can be streamlined in the future.

This technique is not indicated for all distal radius fractures, and simple fractures can be treated with conventional, more cost-effective techniques. However, for complex distal radius fractures in young high-demand patients, this technique might result in more accurate reduction and better outcomes. The results of our patient group are in line with several reports in the literature, with a comparable or even slightly better postoperative range of motion [29,30,31,32,33,34,35,36]. However, we obtained no clinical outcome parameters, had no control group and established our ratings for radiographic outcomes only as either healed or not healed. No further radiographic analysis (e.g., CT scans) was obtained. Therefore, further clinical studies are needed in the future to evaluate the clinical benefit of the technique. This was a feasibility study, and further cases and specific quantification of reduction parameters may yield data supporting the benefits of such computer-assisted trauma surgery.

## 5. Conclusions

The use of patient-specific instruments in a complex distal radius fracture is feasible and facilitates intra-articular reduction without excessive soft tissue dissection. It demonstrates accuracy of reduction, does not delay time to surgery, and results in good patient-reported and functional outcomes without added complications. This may support the further use of the technique in complex multifragmentary distal radius fractures. However, higher costs must be considered for this treatment.

## Figures and Tables

**Figure 1 medicina-58-00744-f001:**
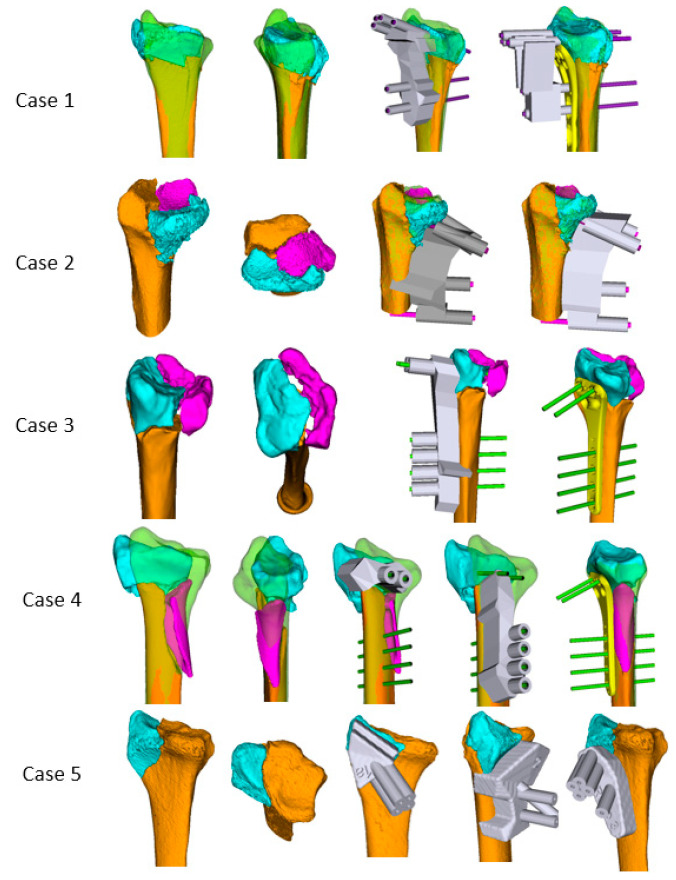
Detailed virtual planning of all five clinical cases (see Table 1 for further details of the cases). The first patient-specific guide facilitates K-wire fixation of separate fragments. Reduction is achieved either with the help of a patient-specific reduction guide, the plate itself, or both.

**Figure 2 medicina-58-00744-f002:**
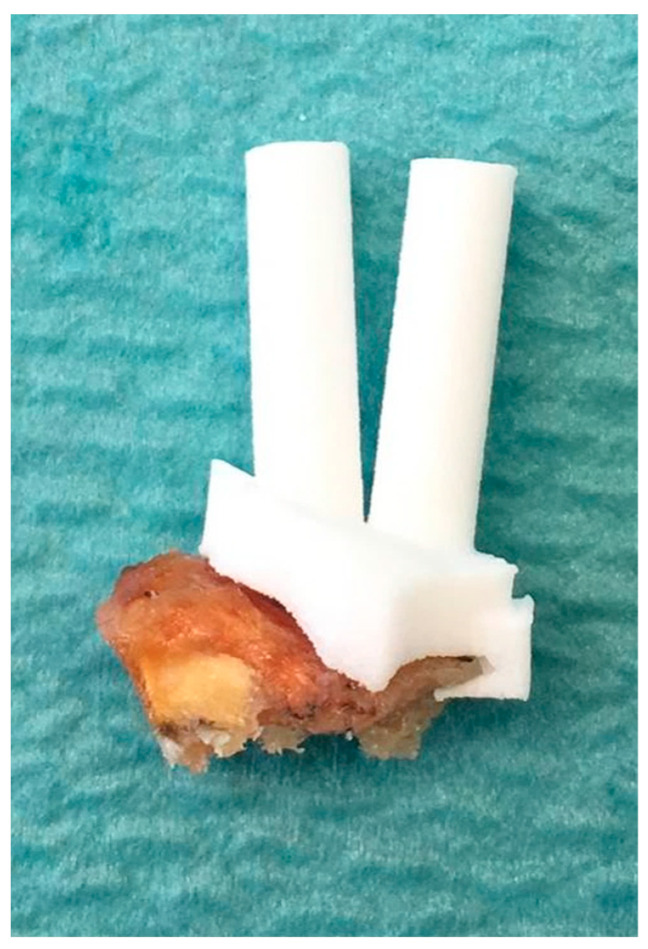
Photograph of a manufactured patient-specific guide positioned on a fracture fragment of a distal radius fracture. The guide will only sit closely on the fragment in the appropriate planned position and allow passing of wires in a predetermined path. Note how the small lip encloses the fracture line.

**Figure 3 medicina-58-00744-f003:**
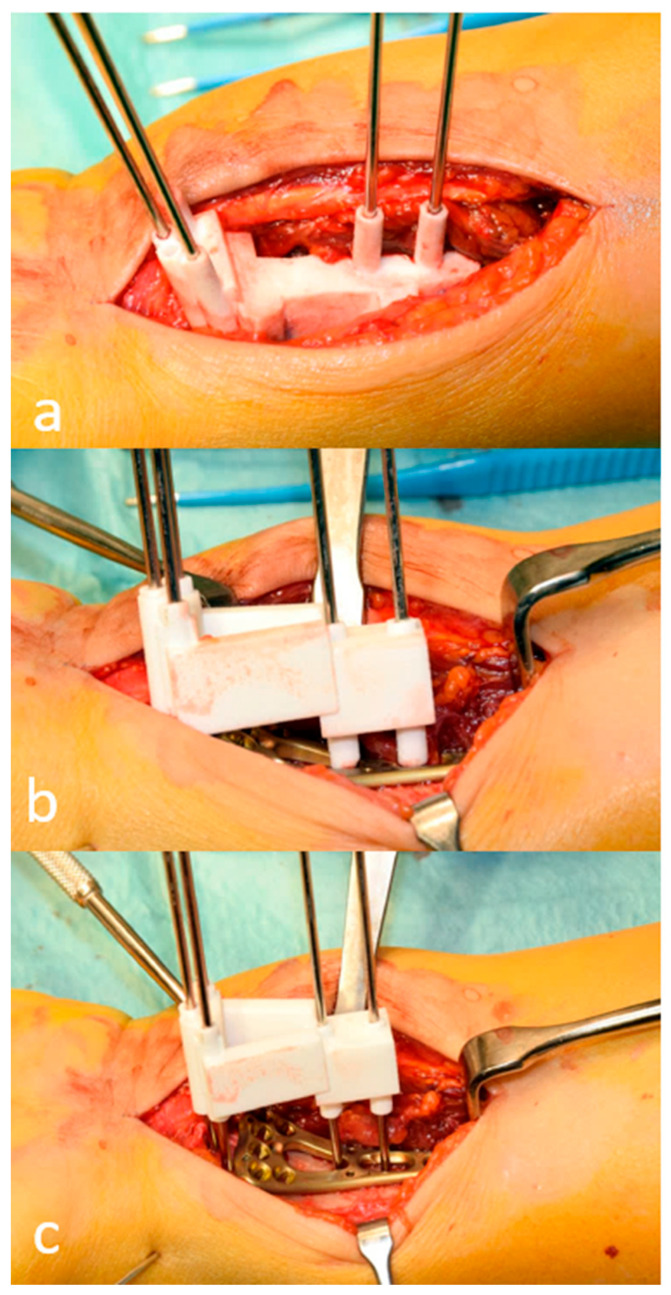
Intraoperative pictures of the first clinical case after a modified Henry approach. (**a**): K-wires are inserted with the help of a patient-specific fragment fixation guide to control reduction later. Note the direction of the K-wires in the distal fracture fragment. (**b**): The first guide is replaced with a second reduction guide which is mounted to a two-column plate (Synthes) and placed over the guide wires. Note how all K-wires are parallel to each other, indicating reduction. (**c**): The guide is removed, and the wires are replaced with locking screws. (A detailed virtual planning of the surgical steps is visualized in Figure 1, Case 1.)

**Table 1 medicina-58-00744-t001:** Detailed overview of the five clinical cases.

#	Age Sex	Fracture Type AO/OTA	Plate (Manufacturer)	Days from Review to Surgery	Back to Work after 3 Months	Healed at 3 Months	Complications
1	50 F	2R3A3.3 Closed	2-collum plate (Synthes)	1	Yes	Yes	None
2	52 M	2R3C1.2 Closed	2-collum plate (Synthes)	10	Yes	Yes	None
3	30 M	2R3C3.2 Closed	Correctus plate (Intercus)	7	Yes	Yes	None
4	64 M	2R3A3.2 Open Grade 1	Correctus plate (Intercus)	7	Yes	Yes	None
5	72 F	2R3B1.1 Closed	2-collum plate (Synthes)	3	Yes	Yes	None

#—case number.

**Table 2 medicina-58-00744-t002:** CT protocol used for all patients in the current study.

**Kv/mAs**	120/120	**Resolution**	Ultra-high
**Thickness**	1.0 (mm)	**Collimation**	20 × 0.625 (mm)
**Increment**	0.5 (mm)	**Pitch**	0.652
**Filter**	Y-Sharp (YE)	**Rotation Time**	0.5 (s)
**Enhancement**	−1.0	**FOV**	150 (mm)
**Windows/Center**	450/2000 (HU)	**I-Dose**	1

**Table 3 medicina-58-00744-t003:** Detailed clinical outcome of the five clinical cases at final follow-up. Values of the healthy contralateral side are presented in brackets.

#	Flexion (°)	Extension (°)	Pronation (°)	Supination (°)	Grip Strength (kg)	Last Follow-Up (Months)
1	75 (75)	70 (70)	70 (70)	80 (80)	38 (65)	6
2	75 (75)	70 (70)	70 (70)	80 (80)	38 (65)	6
3	20 (70)	50 (60)	70 (70)	60 (80)	38 (45)	8
4	50 (70)	40 (50)	75 (75)	65 (65)	32 (36)	12
5	70 (70)	80 (80)	80 (80)	85 (85)	34 (34)	12

#—case number.

## Data Availability

All data relevant to the study are included in the article.

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
