# Peer review of "Three-Dimensional Planning and Patient-Specific Instrumentation for the Fixation of Distal Radius Fractures"

_medicina, 2022, doi:10.3390/medicina58060744_

Round 1

Reviewer 1 Report

MANUSCRIPT TITLE 3D Planning and Patient Specific Instrumentation for the Fixation of Distal Radius Fractures

MANUSCRIPT NUMBER medicina-1695585

REVIEW SUBMISSION DATE 25 April 2022

General comments------------------------------------------------------------------------------------------------------------

The authors presented a brief case series of patients with distal radius fractures, treated using patient-specific 3D-printed instruments. While I don’t have any major concerns about the content of the paper, it should be extensively rewritten as its title is not matching the content. Since the authors were able to include as little as five patients without a control group, the focus of the paper should be put, in my opinion, on the actual planning and manufacturing of the guides. I suggest including the following:

  • - Describe CT scanning in more detail: intensity, filter type, voxel size, reconstruction kernel in
    general. This is important as the imaging will define the final accuracy of your instruments.
    - Were your software packages Materialise and CASPA licensed for clinical use and if not, how
    were you still able to use it for your patients?
    - The same goes for your printer, the printing process and the printing material – was it
    licensed for clinical use? Is this a poly-jet printer, what’s its resolution and printing speed?
    - How did you clean your models? How did you deal with potential residual material that could
    end up in the patient? How did you make sure that there is no alteration in size and/or
    geometry following the sterilization?
    - What exactly was included in the costs ranging 150 – 250€? How much time does it take to
    plan the PSI and what manpower/qualification is required? How long did the planning, and
    printing last and could you estimate the costs.

Besides that the discussion should be extended and concentrate more on your own results, e.g. was the clinical outcome good, compared to conventional surgeries and surgeries using digital planning.
Screen the literature!

Specific comments-------------------------------------------------------------------------------------------------------------

P L Comment

1 1 The abstract does not include any quantitative results. This should be changed.
1 30 It’s “Jupiter et al.” and also the authors' names should be followed by the specific
reference/s. None of the references [1-3] is by Jupiter et al.
1 44 It’s “computer-assisted” (not “computer assisted”)
2 64 You have a double space at the end of the sentence, I think (also in p 7 l 216, p 8 l 237.
O 8 l 241)).
2 91 Add a version of the SW and the city of the manufacturer
3 106 “The PSI guides are two types” – this sentence is odd.
4 129 “Henry” not “henry”
5 136 The images are labelled a-c and the Figures A-C.
6 145f The patient characteristics, their fractures and preoperative planning are not the
results of your study, therefore, they belong to the Materials and Methods section.
7 212 It’s “K-wire”, not “k wire”

Author Response

Review Report, Reviewer 1

The authors presented a brief case series of patients with distal radius fractures, treated using patient-specific 3D-printed instruments. While I don’t have any major concerns about the content of the paper, it should be extensively rewritten as its title is not matching the content. Since the authors were able to include as little as five patients without a control group, the focus of the paper should be put, in my opinion, on the actual planning and manufacturing of the guides. I suggest including the following:

Thank you very much for your supportive review. The intention of this paper was indeed to focus on the actual planning and manufacturing process as we aimed to publish it in the special issue " Digital Trends in the Field of Orthopaedic Implant Research and Development". Therefore, the clinical outcome of the "only" five included patients shift more into the background. However, we wanted to present the first clinical patients ever treated with this new technology to demonstrate its clinical applicability. A case series with the first 20 included patients is currently conducted focusing more on the clinical outcome and the accuracy of this technique compared to conventional treated patients.

1- Describe CT scanning in more detail: intensity, filter type, voxel size, reconstruction kernel in general. This is important as the imaging will define the final accuracy of your instruments.

We added a new table (see below) with a detailed description of the of the used CT Protocol

Table 1: CT protocol used for all patients in the current study

Kv / mAs

120 / 120

Resolution

Ultra-high

Thickness

1.0

Collimation

20 x 0.625

Increment

0.5

Pitch

0.652

Filter

Y-Sharp (YE)

Rotation Time

0.5

Enhancement

-1.0

FOV

150

Windows / Center

450 / 2000

Dose Right

No

I-Dose

1

2- Were your software packages Materialise and CASPA licensed for clinical use and if not, how

were you still able to use it for your patients?

The pipeline from acquisition of the CT-Scan to the application of the patient-specific guides intraoperatively is CE-certified as medical product (MyOsteotomy, Medacta SA, Switzerland). The extension of application area to acute distal radius fractures was ensured within the framework of an ethics application to the local committee.

3- The same goes for your printer, the printing process and the printing material – was it

licensed for clinical use? Is this a poly-jet printer, what’s its resolution and printing speed?

We have received regulatory approval to use the printer for the purpose of the study. General licensing was planned as a subsequent step that is usually outsourced to a MedTech company. The printing material is PA2200 is the commonly used material for patient-specific instruments. It was validated as a class VI (highest class) by the United States Pharmacopeia (USP).

4- How did you clean your models? How did you deal with potential residual material that could

end up in the patient? How did you make sure that there is no alteration in size and/or

geometry following the sterilization?

We followed and showed equivalency of the guide geometry, printing, and cleaning and sterilization process with another CE- and FDA-certified patient-specific instrument on the market. The necessary documentation was sent to the regulatory body as well.

5- What exactly was included in the costs ranging 150 – 250€?

These costs represent only the internal material costs of the printing process.

  1. How much time does it take to plan the PSI and what manpower/qualification is required?

Surgeons and engineers who have expertise in 3D planning of corrective osteotomies performed the planning. A simple distal radius fracture with 1 fragment takes 2-3 hours.

7.How long did the planning, and printing last and could you estimate the costs.

The duration of the printing is dependent on the number of needed parts and lasted several hours including a cooling time of the printer. A printing time of 8-12 hours is estimated for a distal radius fracture.

8 .Besides that the discussion should be extended and concentrate more on your own results, e.g. was the clinical outcome good, compared to conventional surgeries and surgeries using digital planning.

Screen the literature!

As mentioned above the currents study focusses more the actual procedure of 3d PSI and less on the clinical outcomes. Furthermore, no clinical outcome scores have been obtained, thus, comparing the patients of the current study to conventionally treated patients is not really meaningful. However, we added a section to the discussion and compared accuracy of this technique in osteotomies after misshealed distal radius fractures.

Further studies are definitely needed to evaluate the clinical outcome of patients treated with 3d PSI in distal radius or even other fractures.

P L Comment

  1. 1 1 The abstract does not include any quantitative results. This should be changed.

Agreed. We added a section concerning the clinical outcome to the abstract

  1. 1 30 It’s “Jupiter et al.” and also the authors' names should be followed by the specific

reference/s. None of the references [1-3] is by Jupiter et al.

Agreed. The sentence was rephrased.

  1. 1 44 It’s “computer-assisted” (not “computer assisted”)

Corrected

  1. 2 64 You have a double space at the end of the sentence, I think (also in p 7 l 216, p 8 l 237.

O 8 l 241)).

Corrected. Thank you

  1. 2 91 Add a version of the SW and the city of the manufacturer

(Mimics Vers.20.0, Materialise, Leuven, Belgium)

  1. 3 106 “The PSI guides are two types” – this sentence is odd.

Agreed thank you. We rephrased the according section. It now reads: There are two types of PSI guides both of which are needed to reduce the fracture

  1. 4 129 “Henry” not “henry”

Corrected. Thank you

  1. 5 136 The images are labelled a-c and the Figures A-C.

Corrected. Thank you

  1. 6 145f The patient characteristics, their fractures and preoperative planning are not the

results of your study, therefore, they belong to the Materials and Methods section.

Agreed. Thank you. These passages were moved to the M&M section

  1. 7 212 It’s “K-wire”, not “k wire”

Corrected. Thank you

Reviewer 2 Report

Congratulations for this work and nice documented paper. 

Very helpful is the preoperative planning and 3D guided model. 

Absolutely that way is the future and help for the hand surgeons all over the world. 

Author Response

Review Report, Reviewer 2

Congratulations for this work and nice documented paper. 

Very helpful is the preoperative planning and 3D guided model. 

Absolutely that way is the future and help for the hand surgeons all over the world. 

Thank you very much

Round 2

Reviewer 1 Report

General comments:

The authors presented a brief case series of patients with distal radius fractures, treated using patient-specific 3D-printed instruments. There was some improvement in the revised version, however, not all points (in general the authors’ replies to the reviewer 3 – 7 need to be worked out and integrated into the manuscript) have been addressed.
- No information was provided on the printing and cleaning process (see my 1st review).
- Report (in the manuscript!) what exactly was included in the costs ranging 150 – 250€ and the time frame to plan the PSI and what manpower/qualification is required? The costs and possible benefits outweighing these costs should be discussed. The higher costs, when using 3D printing are considered a disadvantage in clinical applications.
- Since no control group was included in the manuscript, the authors could report if the values in Table 3 are in general in accordance/better/worse than what is achieved in classic treatment without PSI (data from literature).

Specific comments:

P  L         Comment
1 22        Please ass the standard deviations to your reported means
1 23        Supination is a German term, that does not exist in English to my knowledge. I'm sorry, I haven't pointed this out during my first review
1 32        The authors are referring to a landmark paper, but cite three. This should be addressed again.
3 116      Table 1, instead of Table 2
6 180      Table 2, instead of 1. Also, this table is nowhere referred to in the manuscript. Please check all tables and their reference in the text.
6 181      Table 1 lacks units. Specify the quantity and the unit in brackets. I'm not an expert, but check if "dose right" is something that is relevant and should be included - otherwise delete.
9 245       It's "kg", not "Kg"
10 364     I would rather use "impact" instead of "force".
10 365     What is to be understood under "normal" landmarks? Furthermore, I believe the right expression is "bone landmarks".

Author Response

Review Report, Reviewer 1 (Round 2)

The authors presented a brief case series of patients with distal radius fractures, treated using patient-specific 3D-printed instruments. There was some improvement in the revised version, however, not all points (in general the authors’ replies to the reviewer 3 – 7 need to be worked out and integrated into the manuscript) have been addressed.
- No information was provided on the printing and cleaning process (see my 1st review).
- Report (in the manuscript!) what exactly was included in the costs ranging 150 – 250€ and the time frame to plan the PSI and what manpower/qualification is required? The costs and possible benefits outweighing these costs should be discussed. The higher costs, when using 3D printing are considered a disadvantage in clinical applications.
- Since no control group was included in the manuscript, the authors could report if the values in Table 3 are in general in accordance/better/worse than what is achieved in classic treatment without PSI (data from literature).

Thank you again for your very supportive review. Please find below our revised answers to the questions 3-7 from round 1 of your review and our answers to round 2

 Review Report, Reviewer 1 (Round 1):

 3- The same goes for your printer, the printing process and the printing material – was it

licensed for clinical use? Is this a poly-jet printer, what’s its resolution and printing speed?

We added the following section to the manuscript:

We have received regulatory approval to use the printer for the purpose of the study. General licensing was planned as a subsequent step that is usually outsourced to a MedTech company. The printing material is PA2200, which is the commonly used material for patient-specific instruments. It was validated as a class VI (highest class) by the United States Pharmacopeia (USP). The extreme fine focus diameter of our inhouse printer (Formiga P100, EOS GmbH, Krailling, Germany) enables wall thickness of 0.4mm at a slice thickness of 100 µm and a printing speed of 24 mm height/h to be build.

4- How did you clean your models? How did you deal with potential residual material that could

end up in the patient? How did you make sure that there is no alteration in size and/or

geometry following the sterilization?

 - No information was provided on the printing and cleaning process (see my 1st review).

Agreed. We added the following sentence to the manuscript.

Cleaning and sterilization was performed according to the "cleaning and sterilization instructions of the MyOsteotomy (Medacta SA, Castel San Pietro, Switzerland). We showed equivalency to this approved medical product and the necessary documentation was provided to regulatories.

 5- What exactly was included in the costs ranging 150 – 250€?

 These costs represent only the internal material costs of the printing process.

- Report (in the manuscript!) what exactly was included in the costs ranging 150 – 250€ and the time frame to plan the PSI and what manpower/qualification is required?  The costs and possible benefits outweighing these costs should be discussed. The higher costs, when using 3D printing are considered a disadvantage in clinical applications.

Thank you for pointing out this important issue again. We discussed this intensively with our in-house engineers and added a new section to the manuscript. You are absolutely right. The overall costs are higher as the reported 150-250€ which are only the material costs of the printing process. It is difficult to precisely quantify the planning production costs of these experimental guides. However, MyOsteotomy is a spinoff of our research campus which was bought by Medacta. We do know the costs of the MyOsteotomy guides: Planning and design (1500€) Production (600€)

new section in the conclusion: However, higher costs must be considered for the treatment. 

new section in M&M: Overall costs were around 1500€ for planning and design of the PSI as well as 600€ for production of the PSI including material costs of around 200€.

 new section in the discussion (limitations of this technique): Furthermore, planning, design and production of the PSI are expensive and the time requirement from a surgeon, as well as an available engineer with software access is not negligible. An ongoing critique of many examples of computer assisted orthopedic surgery is that the time and cost investment are currently disproportionate to the demonstrated benefits.

  1. How much time does it take to plan the PSI and what manpower/qualification is required?

 We added this to the manuscript: Surgeons and engineers who have expertise in 3D planning of corrective osteotomies performed the planning, which took around 2-3 hours.

 7.How long did the planning, and printing last and could you estimate the costs.

 We added this to the manuskript

The duration of the printing is dependent on the number of needed parts and lasted several hours including a cooling time of the printer. A printing time of 8-12 hours is estimated for a distal radius fracture.

Review Report, Reviewer 1 (Round 2):

Specific comments:

P  L         Comment
1 22        Please add the standard deviations to your reported means

agreed. We changed this in the abstract

1 23        Supination is a German term, that does not exist in English to my knowledge. I'm sorry, I haven't pointed this out during my first review.

We checked this with our native speaking co-workers, and they agreed that pro- and supination are correct terms
1 32        The authors are referring to a landmark paper but cite three. This should be addressed again.

agreed. thank you for pointing out this issue again. We rephrased the sentence to several authors suggested …
3 116      Table 1, instead of Table 2

We checked all Tables and Figures in the manuscript.

6 180      Table 2, instead of 1. Also, this table is nowhere referred to in the manuscript. Please check all tables and their reference in the text.

We checked all Tables and Figures in the manuscript.

6 181      Table 1 lacks units. Specify the quantity and the unit in brackets. I'm not an expert, but check if "dose right" is something that is relevant and should be included - otherwise delete.

Thank you for pointing this out. Units were added. We checked again with our radiology department and removed dose right from the table. This is an irrelevant information.

9 245       It's "kg", not "Kg"

thank you
10 364     I would rather use "impact" instead of "force".

agreed. We changed this in the manuscript

10 365     What is to be understood under "normal" landmarks? Furthermore, I believe the right expression is "bone landmarks".

agreed. We changed the according section.

- Since no control group was included in the manuscript, the authors could report if the values in Table 3 are in general in accordance/better/worse than what is achieved in classic treatment without PSI (data from literature).

 Agreed. Although, this is very difficult to compare our patients to the current literature as we obtained no clinical outcome parameters. Different postoperative protocols, operation techniques, surgical experience as well as different fracture types need to be considered. However, leaving all these aspects aside our results are comparable or even slightly better than reports in the current literature. A section was added to the discussion pointing out that further clinical research is needed to evaluate the new technique in terms of clinical and radiographic outcome.

Round 3

Reviewer 1 Report

Thank you for the revision. I don't have any further comments.